# Residual Effect of Organic and Inorganic Fertilizers on Growth, Yield and Nutrient Uptake in Wheat under a Basmati Rice–Wheat Cropping System in North-Western India

**Salwinder Singh Dhaliwal** [1], **Vivek Sharma** [1], **Arvind Kumar Shukla** [2,*], **Rajeev Kumar Gupta** [1], **Vibha Verma** [3], **Manmeet Kaur** [3], **Sanjib Kumar Behera** [2] **and Prabhjot Singh** [1]

1   Department of Soil Science, Punjab Agricultural University, Ludhiana 141004, India
2   ICAR-Indian Institute of Soil Science, Bhopal 462038, India
3   Department of Chemistry, Punjab Agricultural University, Ludhiana 141004, India
*   Correspondence: arvindshukla2k3@yahoo.co.in

**Abstract:** Restoring soil fertility in farming systems is essential to sustain a crop and its productivity. Thus, the present study was conducted to assess the residual effects of the combined application of fertilizers and manures on yield, concentration and uptake of nutrients in wheat under basmati rice-wheat cropping system. The treatments applied in the present study involve T1: control, T2: farmyard manure (15 t ha$^{-1}$), T3: poultry manure (6 t ha$^{-1}$), T4: press mud (15 t ha$^{-1}$), T5: rice straw compost (6 t ha$^{-1}$), T6: farmyard manure (15 t ha$^{-1}$) + 50% recommended dose of nitrogen (RDN), T7: poultry manure (6 t ha$^{-1}$) + 50% RDN, T8: press mud (15 t ha$^{-1}$) + 50% RDN, T9: rice straw compost (6 t ha$^{-1}$) + 50% RDN, T10: 75% RDN, T11: farmyard manure (15 t ha$^{1}$) + 75% RDN, T12: poultry manure (6 t ha$^{-1}$) + 75% RDN, T13: press mud (15 t ha$^{-1}$) + 75% RDN, T14: rice straw compost (6 t ha$^{-1}$) + 75% RDN, T15: 100% RDN. The residual effects of organic manures significantly improved the growth parameters and yield attributes. Among different residual treatments, the use of farmyard manure + 75% NPK showed maximum plant height (125.2 cm), number of tillers (68.0 m$^{-1}$), chlorophyll content (45.0) and yield (50.84 q ha$^{-1}$ for grain and 80.43 q ha$^{-1}$ for straw, respectively). Additionally, the incorporation of farmyard manure + 75% RDN demonstrated the highest uptake of nitrogen, phosphorus and potassium in grain (7.37, 3.31 and 4.93 g ha$^{-1}$, respectively) and straw (1.72, 1.05 and 12.63 g ha$^{-1}$, respectively). The maximum zinc, copper, iron and manganese concentrations were observed to be 32.0, 3.1, 52.1 and 17.6 mg kg$^{-1}$ in grain and 8.2, 2.1, 374.6 and 20.5 mg kg$^{-1}$ in straw, respectively. Similarly, the highest uptakes were observed to be 67.6, 15.5, 263.8 and 89.6 g ha$^{-1}$ in grain and 173.3, 16.8, 3026.9 and 170.6 g ha$^{-1}$ in straw, respectively. Thus, the integrated application of farmyard manure with 75% RDN could be used to sustain wheat productivity and maintain soil fertility which otherwise deteriorates due to the sole application of inorganic fertilizers.

**Keywords:** wheat; manures; organic and inorganic fertilizers; yield; uptake

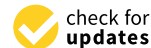



## 1. Introduction

Wheat (*Triticum aestivum* L.) is the most widely grown cereal crop in terms of area as well as productivity, and it feeds around two-thirds of the world's population [1,2]. However, lower yield outcomes have been recorded due to poor soil conditions, low quality seeds, late harvesting and planting, uneven fertilizer application and a lack of irrigation water [3,4]. Moreover, degradation of soil due to remarkable enhancement in population and urbanization, along with lack of quality of seed, have resulted in food insecurity in emerging nations [5–7]. Additionally, the overuse of inorganic fertilizers to overcome the demand for food supplies often causes N losses through several pathways, including ammonia volatilization, leaching and greenhouse gas emission, major environmental problems and overall N balancing [8]. As a result, new agronomic technologies have been adopted to increase the efficiency of fertilizers and to decrease the nitrogen losses through

leaching and volatilization which are extremely important for sustainable agriculture. At present, the use of organic manures from different animals (pig and cow), biochar and crop residues directly or in the form of mixtures is widely investigated by researchers. Organic manures help in maintaining soil fertility by providing important plant nutrients, particularly micronutrients, which increase crop growth [9]. Use of manures also improves the quality of soil physically, chemically and biologically [10], thus having a long-term impact on subsequent crops [11,12]. Green manure (*Sesbania aculeata*), farmyard manure, press mud (a sugar waste product) and wheat residue are key sources of organic matter that can improve soil fertility and help plants utilize fertilizers more efficiently. Farmyard manure improves soil structure, water and nutrient retention capacity, root penetration, microbial activity and nutrient exchange capacity, which, in turn, enhances the fertility of soil [13,14]. Additionally, biochar can be used as a soil amendment, promoting nutrient supply and reducing greenhouse gas emissions [15,16]. Additionally, the partial replacement of manures with inorganic fertilizers would reduce the production cost to a greater level [17]. Previous studies have reported that substitution of chemical fertilizers with organic manure has beneficial effects on crop yield [18]. Use of different manures (press mud, poultry manure and farmyard manure) in rice–wheat systems have been evidenced to increase the crop yield and DTPA-extractable micronutrient (Zn, Cu, Fe and Mn) concentrations in soil in north-western India [19]. The improvement in growth, along with yield attributes of rice and wheat through the combined use of organic and inorganic fertilizers, has been reported by Sarwar et al. [20]. In a long-term field experiment of 40 years of soybean–maize rotation, application of organic manures on brown soil improved the crop yields along with soil properties [21]. Additionally, the use of farmyard manure is favored over other organic fertilizers due to its faster disintegration time as compared to other sources.

Apart from organic manures, residue retention has also been promoted as a viable option for soil amendment to increase the crop yield, while lowering the use of chemical fertilizers. With the increase in crop production, the amount of crop residue produced per year has also increased. Returning crop residue to the soil helps to improve the organic matter content and soil physico-chemical and biological properties, thus preserving the soil fertility [22]. The existing literature has shown positive impacts of crop residue retention on crop yield. For instance, adding rice residue to a *Typic Ustochrept* sandy loam soil in north-western India enhanced the wheat yield considerably when compared to untreated plots [23]. The retention of crop residue in the rice–wheat cropping system has boosted the micronutrient content of the crop [19].

To the best of our knowledge, information on the effects of different combinations of organic and chemical fertilizers on growth and yield attributes in wheat is still lacking. Therefore, the present study was conducted with the objective to examine the effect of different fertilizer management practices involving the combination of organic manures, i.e., farmyard, poultry, press mud and rice straw compost/crop residue along with NPK on growth parameters, yield and concentration, along with nutrient uptake in wheat.

## 2. Materials and Methods

### 2.1. Site Specification and Characteristics

The present study was conducted during two consecutive *Rabi* seasons, of 2019–2020 and 2020–2021, at the experimental farm, Department of Soil Science, Punjab Agricultural University, Ludhiana, Punjab (30°56′ N, 75°52′ E and 247 m above mean sea level) on the Indo-Gangetic plains of north-western India. The climate in the region is subtropical, with hot and rainy summers along with dry winters. The months from July to September receive most of the rainfall, around 70%, with rainfall of 400–600 mm annually. The total rainfall during the crop season from October to April was 219 and 68.9 mm during 2019–2020 and 2020–2021, respectively The average monthly maximum temperature of the study area varied from 15.9 °C to 32.8 °C during 2019–2020 and 16.4 °C to 34.2 °C during 2020–2021, respectively, however, the minimum temperature varied from 6.7 °C to 18.4 °C during

2019–2020 and 7.1 °C to 17.0 °C during 2020–2021, respectively, during the growing season of wheat (Figure 1).

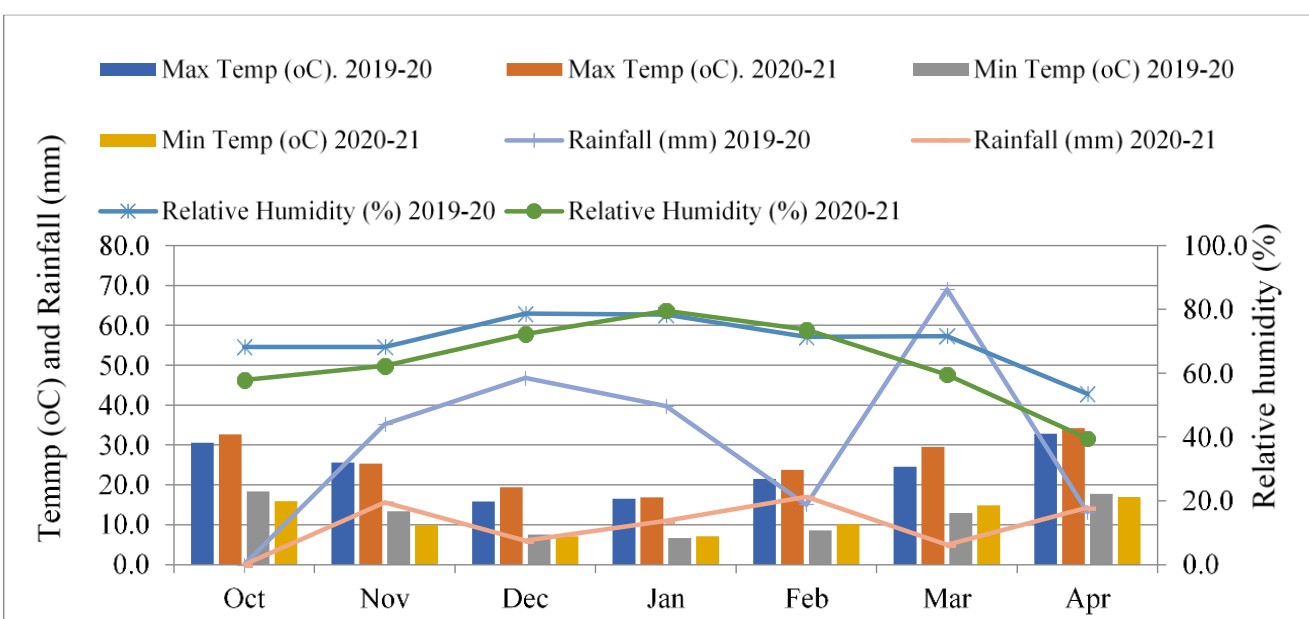

**Figure 1.** Monthly average maximum and minimum temperature, relative humidity and rainfall of the study area.

*2.2. Treatment Details*

The study was carried out by using 15 treatments (Table 1) in a randomized block design with three replications. The manures used in the present study were farmyard manure (FYM 15 t ha$^{-1}$), poultry manure (PM 6 t ha$^{-1}$), press mud (PrM 15 t ha$^{-1}$) and rice straw residue (RSC 6 t ha$^{-1}$) as per the recommended dose of Punjab Agricultural University, Ludhiana [24]. The organic manures were added to the soil once a year only before transplanting of basmati rice, and their residual effects were examined on wheat crops. Among the different treatments, only the nitrogen level varied per treatment, however the phosphorus level was same for all treatments.

Basmati variety 'Punjab Basmati 5' was transplanted during the second week of July. The recommended dose of nitrogen (RDN) for basmati rice was 40 Kg ha$^{-1}$. The sowing of wheat variety PBW 725 was performed in November with a row to row spacing of 15–20 cm and plot size of 6.0 × 1.2 m$^2$. The wheat crop was sown with a seed-cum-fertilizer drill method at a depth of 4–6 cm. The recommended doses of N (125 kg ha$^{-1}$) and P$_2$O$_5$ (62.5 kg ha$^{-1}$) were applied to the wheat crop. A half dose of N and full dose of P were applied at the time of sowing, whereas the remaining N was divided into two doses, i.e., at first and second irrigation. Additionally, the wheat yield was calculated from the net plot area, leaving border rows, and was later changed to q ha$^{-1}$. The soil in the present experiment was sandy loam with pH, EC and OC of 6.80, 0.16 dSm$^{-1}$ and 0.19%, respectively [25–27]. The method by Subbiah and Asija, the Olsen extractable P method and the method by Merwin and Peech were used to estimate the availability of N, P, and K to be 153 kg ha$^{-1}$, 23.9 kg ha$^{-1}$ and 138 kg ha$^{-1}$ in soil, respectively [28–30]. The initial DTPA extractable micronutrient levels, i.e., Zn, Cu, Fe and Mn in soil were 0.46 mg kg$^{-1}$, 0.28 mg kg$^{-1}$, 1.42 mg kg$^{-1}$ and 4.19 mg kg$^{-1}$, respectively [31], as calculated using the instrument AAS (Varian AAS-FS 240 model, Burladingen, Germany).

**Table 1.** Treatment details along with the N added in each treatment through manures and fertilizers.

| Sr No | Treatments | Rice | | Wheat | |
|---|---|---|---|---|---|
| | | Manure-N | Fertilizer-N | Manure-N | Fertilizer-N |
| $T_1$ | Control (No fertilizers) | - | - | - | - |
| $T_2$ | Farmyard manure (15 t ha$^{-1}$) | 90 | - | - | - |
| $T_3$ | Poultry manure (6 t ha$^{-1}$) | 72 | - | - | - |
| $T_4$ | Press mud (15 t ha$^{-1}$) | 120 | - | - | - |
| $T_5$ | Rice straw compost (6 t ha$^{-1}$) | 30 | - | - | - |
| $T_6$ | Farmyard manure (15 t ha$^{-1}$) + 50% RDN | 90 | 20 | - | 62 |
| $T_7$ | Poultry manure (6 t ha$^{-1}$) + 50% RDN | 72 | 20 | - | 62 |
| $T_8$ | Press mud (15 t ha$^{-1}$) + 50% RDN | 120 | 20 | - | 62 |
| $T_9$ | Rice straw compost (6 t ha$^{-1}$) + 50 % RDN | 30 | 20 | - | 62 |
| $T_{10}$ | 75% Recommended fertilizers RDN | - | 30 | - | 93 |
| $T_{11}$ | Farmyard manure (15 t ha$^{-1}$) + 75% RDN | 90 | 30 | - | 93 |
| $T_{12}$ | Poultry manure (6 t ha$^{-1}$) + 75% RDN | 72 | 30 | - | 93 |
| $T_{13}$ | Press mud (15 t ha$^{-1}$) + 75% RDN | 120 | 30 | - | 93 |
| $T_{14}$ | Rice straw compost (6 t ha$^{-1}$) + 75% RDN | 30 | 30 | - | 93 |
| $T_{15}$ | 100% Recommended fertilizers RDN | - | 40 | - | 125 |

Farmyard manure (0.6%N), Poultry manure (1.2%N), Press mud (0.8%N), Rice straw compost (0.5%N).

### 2.3. Harvesting and Analysis

The height of plant, total number of tillers per meter row length and chlorophyll content through chlorophyll meter (SPAD 502) were noted. When the crop reached physiological maturity, it was manually harvested, and grain as well as straw samples were collected for examination. To measure the dry weight of different components of plant, the samples were air-dried before drying in an oven at 65 °C for 48 hrs. A mechanical grinder was used to grind oven dried plant samples to a fine powder. On an electric hot plate, ground samples of grain and straw weighing 1.0 g each were subjected to digestion using a mixture of di-acid, i.e., $HNO_3$ and $HClO_4$ acid in a 3:1 ratio [32]. The micronutrient content (Zn, Fe, Mn and Cu) in digested extracts of plant was measured using an atomic absorption spectrophotometer (Model AAS 240 FS, Company Varian, Germany). The digested and filtered samples were further examined for total P content through the vanadomolybdo-phosphoric-yellow color technique in a system containing nitric acid [33] using a spectrophotometer. The total K as well as micronutrient content was observed through flame photometric [34] and atomic absorption spectrophotometric detection [35], respectively. The samples were analyzed for N using Kjeldahl's method [36]. The uptake of nutrients was assessed by multiplication of concentrations and respective yields, as shown in Equation (1).

$$\text{Macronutrient uptake} \left( \text{kg ha}^{-1} \right) = \frac{\text{Concentration (\%)} \times \text{Yield} \left( \text{q ha}^{-1} \right)}{10} \quad (1)$$

$$\textit{Micronutrient uptake} \left( \text{g ha}^{-1} \right) = \frac{\textit{Concentration} \left( \text{mg kg}^{-1} \right) \times \textit{Yield} \left( \text{q ha}^{-1} \right)}{10} \quad (2)$$

### 2.4. Statistical Analysis

Data were analysed statistically using SPSS version 16.0 (SPSS Inc., Chicago, IL, USA) packages. One-way analysis of variance (ANOVA) and Duncan Multiple Range test were

performed in order to assess least significant difference (LSD) at a probability of 0.05 between the treatment results on the crop.

## 3. Results and Discussion

The results of the study support the hypothesis that crop residue would improve the crop attributes, yield and macro- and micro-nutrient uptake in the grain and straw of wheat. The various parameters analysed in the present study are discussed in the following sections.

### 3.1. Impact of Manures and Fertilizers Application on Plant Growth and Yield Attributes of Wheat

Sole as well as combined use of organic manures and inorganic fertilizers exhibited significant effects on various growth parameters, as given in Table 2. The data displayed that the height of the plant, number of tillers and chlorophyll content ranged from 73.1 to 125.2 cm, 48 to 68 $m^{-1}$ and 28.1 to 45.0, respectively. The maximum plant height was observed in treatment T11, which was significantly higher (71.3%) than T1. The results of treatment T11 were statistically on par with treatments T6, T12 and T14. Significantly, lower plant height in treatment T1 as compared to rest of the treatments clearly indicated the role of farmyard manure + 75% RDN in enhancing the growth parameters. In context of the number of productive tillers, the effect of manure and fertilizer application was non-significant. The chlorophyll content varied significantly among the different treatments, with the maximum content observed in treatment T11, which was significantly higher (60.1%) than the control. Additionally, the result of T11 was statistically on par with treatments T12, T13, T14 and T15.

**Table 2.** Effect of different fertilizers and manures on plant growth and yield attributes of wheat.

| Treatments | Plant Height (cm) | No. of Productive Tillers per Meter Row Length | Chlorophyll Content |
|---|---|---|---|
| T$_1$ | 73.1 [e] | 48 | 28.1 [f] |
| T$_2$ | 109.3 [bcd] | 55 | 38.3 [cd] |
| T$_3$ | 102.9 [cd] | 53 | 34.4 [de] |
| T$_4$ | 96.8 [d] | 50 | 33.2 [e] |
| T$_5$ | 100.8 [d] | 51 | 33.1 [e] |
| T$_6$ | 114.3 [abc] | 63 | 39.9 [bc] |
| T$_7$ | 108.7 [bcd] | 63 | 36.9 [cde] |
| T$_8$ | 97.4 [d] | 54 | 33.1 [e] |
| T$_9$ | 104.4 [cd] | 61 | 39.5 [bc] |
| T$_{10}$ | 103.8 [cd] | 58 | 38.0 [cde] |
| T$_{11}$ | 125.2 [a] | 68 | 45.0 [a] |
| T$_{12}$ | 121.3 [ab] | 66 | 43.9 [ab] |
| T$_{13}$ | 108.9 [bcd] | 62 | 41.0 [abc] |
| T$_{14}$ | 118.4 [ab] | 63 | 42.7 [ab] |
| T$_{15}$ | 108.7 [bcd] | 60 | 41.7 [abc] |
| LSD ($p \leq 0.05$) | 13.2 | NS | 4.9 |

Treatment details are given in Table 1. The values with identical superscript letters do not differ significantly at the 5% level by Duncan's Multiple Range test. NS—nonsignificant.

The higher nitrogen concentration in manures, which influenced plant vegetative growth, may have resulted in the significant effect of various manures on wheat plant height. Additionally, the differences in plant height could have been caused by differences

in manure nitrogen composition [37]. The application of farmyard manure raises soil organic matter, which improves soil structure and increases nutrient availability at the same time [38]. This directly contributes to enhanced crop growth and yield, and indirectly affects the soil physical qualities, i.e., aggregate stability and porosity, which can boost root growth and stimulate plant growth. These readily available soil soluble nutrients are quickly absorbed by plants, increasing the amount of nutrients required for cell development, cell division and elongation of the cell in the meristematic region of the plant, assisting in stem elongation [39,40]. Furthermore, manures supply plants with adequate nutrition, particularly micronutrients, resulting in the development of plants with enhanced numbers of tillers [41]. One of the studies reported that the combined application of farmyard manure (15 t/ha) with 120 kg N/ha significantly improved plant height from 44.5 to 55.7 cm, whereas number of tillers showed an increment from 241 to 529 $m^{-1}$ in wheat under long term experimental conditions [42]. In this study, the use of farmyard manure + 75% RDN resulted in maximum chlorophyll content in wheat. This may be due to the nutrient release by manure and fertilizer towards the post-anthesis stage. Thus, the nutrients were available to develop the site of photosynthesis, thereby aiding yield development of the crop. Additionally, synergistic application of both organic and inorganic fertilizers may provide a sufficient level of nutrition which takes part in biosynthesis of chlorophyll. Chlorophyll biosynthesis is mainly affected by light, temperature, carbohydrate and availability of N. In the present study, the addition of farmyard manure and NPK nutrients increased the growth as well as production of plants because these materials increase the growth of roots, reinforce plant stems and enhance photosynthetic rates. Thus, the available nutrients increased the biosynthesis of chlorophyll and organ formation in leaves [43].

### 3.2. Impact of Manures and Fertilizers Application on Wheat Grain and Straw Yield

The effect of manures on yield of wheat was studied during *Rabi* seasons of 2020 and 2021, and the result of two-year data depicted that the use of manures had a significant influence on the yield of grain and straw in wheat. The mean of two-year data demonstrated that the average wheat yield in grain varied from 33.1 q $ha^{-1}$ to 50.8 q $ha^{-1}$, whereas yield varied from 58.2 q $ha^{-1}$ to 80.9 q $ha^{-1}$ in straw, respectively (Table 3). The integrated application of manures and fertilizers significantly enhanced the wheat yield over control. The treatment T11 showed maximum values of grain and straw, which were significantly higher (53.5% and 39.0%, respectively) than control. Additionally, the application of increased rate of N further improved the wheat yield and thus the treatments involving 75% RDN along with manures were found to have more potential for enhancing the yield. With the same recommended fertilizer doses, farmyard manure-treated crop exhibited enhanced yield compared to other manures. Additionally, in the case of grain yield, T11 was not statistically different from treatments T12 and T14. However, in straw yield, treatment T11 was not statistically different from treatments T12, T14 and T15.

The higher yield may be due fact that these organic manures which supply direct available nutrients such as nitrogen to the plants and these organic manures improves the proportion of water stable aggregates of the soil. This was attributed to the cementing action of polysaccharides and other organic compounds released during the decomposition of organic matters, thus leading to taller plants, increased number of leaves, tillers and in turn the final yield [44]. Further, the enhanced grain and straw yields with the use of farmyard manure and 75% RDN might be attributed to improved soil characteristics and crop productivity [45,46]. Additionally, farmyard manure not only increases N availability, but also delivers micronutrients which enhance P and K use efficiency [47], resulting in improved growth and production attributes. Additionally, farmyard manure increases water holding capacity and minimizes leaching losses, increasing the availability of nitrogen to plants and, as a result, increasing wheat grain and straw yield. Furthermore, the higher output in the poultry manure treatment compared to press mud and rice straw compost could be related to the content of uric acid present in poultry manure, which accelerates the rate of nutrient release from poultry manure. These findings are consistent with those

of Islam et al. [1], who found that applying manures and fertilizers together boosted wheat grain and straw output. Similarly, Asit et al. discovered that applying manures and fertilizers together enhanced the yield in a rice–wheat cropping system [48].

**Table 3.** Effect of different fertilizers and manures on grain and straw yield of wheat.

| Treatments | Wheat Yield (q ha$^{-1}$) | | | | | |
|---|---|---|---|---|---|---|
| | Grain | | | Straw | | |
| | I Year | II Year | Average | I Year | II Year | Average |
| T$_1$ | 32.8 $^{ef}$ | 33.3 $^{h}$ | 33.1 $^{f}$ | 53.8 $^{de}$ | 62.5 $^{d}$ | 58.2 $^{f}$ |
| T$_2$ | 31.9 $^{ef}$ | 42.2 $^{efg}$ | 37.1 $^{ef}$ | 52.8 $^{de}$ | 75.6 $^{abc}$ | 64.1 $^{def}$ |
| T$_3$ | 34.1 $^{df}$ | 39.1 $^{g}$ | 36.6 $^{ef}$ | 48.5 $^{e}$ | 75.5 $^{abc}$ | 62.0 $^{ef}$ |
| T$_4$ | 31.4 $^{f}$ | 39.3 $^{g}$ | 35.4 $^{ef}$ | 52.9 $^{de}$ | 67.8 $^{cd}$ | 60.4 $^{ef}$ |
| T$_5$ | 32.3 $^{ef}$ | 39.7 $^{fg}$ | 35.9 $^{ef}$ | 51.3 $^{e}$ | 69.9 $^{bcd}$ | 60.6 $^{ef}$ |
| T$_6$ | 37.7 $^{cd}$ | 44.3 $^{ef}$ | 41.0 $^{ce}$ | 60.3 $^{cd}$ | 76.7 $^{abc}$ | 68.5 $^{bcde}$ |
| T$_7$ | 36.7 $^{cd}$ | 41.3 $^{efg}$ | 39.0 $^{def}$ | 59.8 $^{cd}$ | 71.7 $^{bcd}$ | 65.7 $^{def}$ |
| T$_8$ | 31.6 $^{ef}$ | 43.2 $^{efg}$ | 37.4 $^{ef}$ | 58.4 $^{cd}$ | 68.8 $^{cd}$ | 63.6 $^{ef}$ |
| T$_9$ | 34.8 $^{def}$ | 42.7 $^{efg}$ | 38.7 $^{def}$ | 58.2 $^{cd}$ | 70.7 $^{bcd}$ | 64.4 $^{def}$ |
| T$_{10}$ | 35.3 $^{de}$ | 45.3 $^{cde}$ | 40.3 $^{cd}$ | 64.5 $^{bc}$ | 69.9 $^{bcd}$ | 67.2 $^{cdef}$ |
| T$_{11}$ | 45.9 $^{a}$ | 55.7 $^{a}$ | 50.8 $^{a}$ | 75.9 $^{a}$ | 85.9 $^{a}$ | 80.9 $^{a}$ |
| T$_{12}$ | 44.5 $^{a}$ | 50.0 $^{bc}$ | 47.3 $^{ab}$ | 74.9 $^{a}$ | 78.9 $^{ab}$ | 76.9 $^{abc}$ |
| T$_{13}$ | 43.9 $^{ab}$ | 44.1 $^{def}$ | 43.9 $^{bcd}$ | 63.6 $^{bc}$ | 72.4 $^{bcd}$ | 68.0 $^{bcdef}$ |
| T$_{14}$ | 39.4 $^{c}$ | 53.0 $^{ab}$ | 46.2 $^{abc}$ | 69.6 $^{ab}$ | 78.5 $^{abc}$ | 74.1 $^{abcd}$ |
| T$_{15}$ | 40.1 $^{bc}$ | 48.6 $^{bcd}$ | 44.4 $^{bcd}$ | 75.2 $^{a}$ | 80.7 $^{ab}$ | 77.9 $^{ab}$ |
| LSD ($p \leq 0.05$) | 3.8 | 4.8 | 5.9 | 8.7 | 10.8 | 10.0 |

Treatment details are given in Table 1. The values with identical superscript letters do not differ significantly at the 5% level by Duncan's Multiple Range test. NS—nonsignificant.

*3.3. Impact of Manures and Fertilizers Application on Macronutrients Concentration of Grain and Straw*

Mean of two-year data for grain and straw N, P and K concentration in wheat due to the use of manures and fertilizers is presented in Figure 2. In grain, the N, P and K concentrations varied from 1.22% to 1.44%, 0.49% to 0.66% and 0.51% to 0.97%, respectively, under different treatments. Similarly in straw, the concentrations of N, P and K varied from 0.09% to 0.20%, 0.50% to 0.65% and 1.06% to 1.53%, respectively. Additionally, treatment T11 possessed maximum N, P and K concentrations in grain, which was significantly higher (18.0%, 34.7% and 90.2%) than control. In straw samples also, treatment T11 possessed maximum and significantly higher N, P and K concentrations (18.0%, 34.7% and 90.2%) than control. Additionally, T11 was not statistically different from treatment T12 for N, P and K concentration in grain. However, in straw yield, treatment T11 was not statistically different from treatment T12 for N concentration.

The increased concentration of macronutrients in wheat might be related to the decomposition of organic matter, which results in the incorporation of macronutrients into the soil matrix, allowing the soil to act as a reservoir of these nutrients [49]. Further, these nutrients are released and become available for uptake by plants. Otherwise, humus, which is the final component of organic matter decomposition, accumulates in the environment to increase the moisture retention and nutrient supply potentials of soil. The obtained data were confirmed by the results found by Khater et al. [50], who mentioned that farmyard manure plays an important role in supplying important nutrients required by plants.

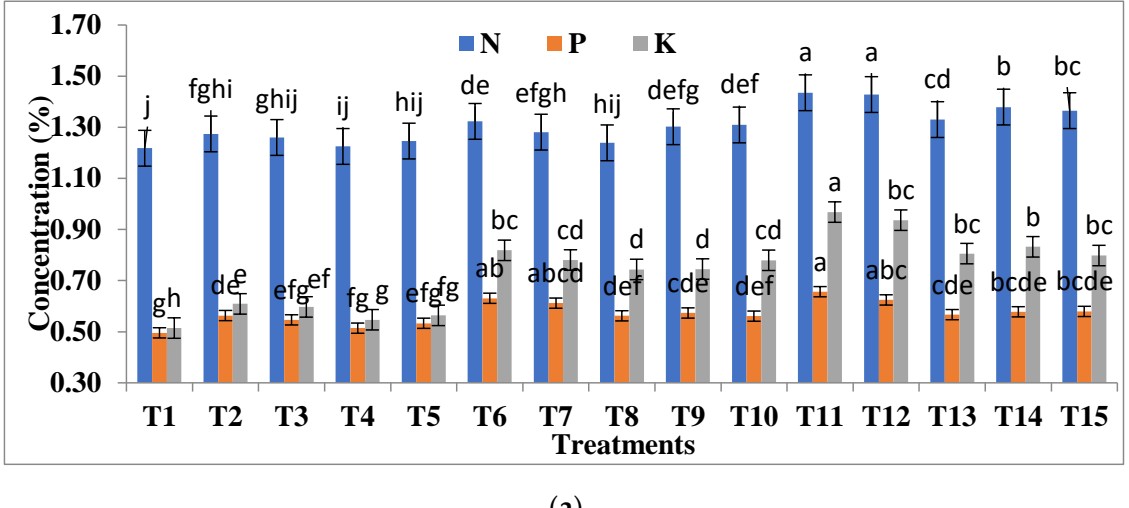

(**a**)

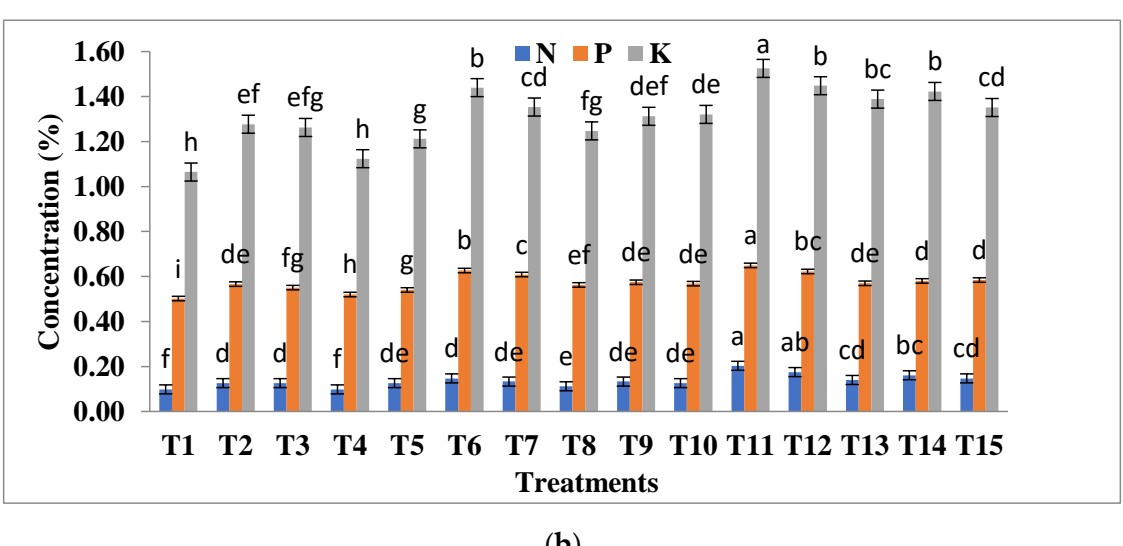

(**b**)

**Figure 2.** Effect of different fertilizers and manures on macronutrient concentration (N, P and K) in (**a**) grain and (**b**) straw of wheat. Treatment details are given in Table 1. The bar with similar or dissimilar letter(s) was evaluated with the least significant difference (LSD) multiple range tests using a probability level of $p \leq 0.05$.

*3.4. Impact of Manures and Fertilizers Application on Micronutrients Concentration of Grain and Straw*

Data for Zn, Cu, Fe and Mn concentration in grain and straw of wheat due to the use of manures and fertilizers are presented in Figure 3. In grain, the Zn, Cu, Fe and Mn concentration increased up to 46.1%, 93.8% mg kg$^{-1}$, 53.7% and 47.9%, respectively, under different treatments. Similarly, in straw, the concentration of Zn, Cu, Fe and Mn varied up to 38.9%, 133.3%, 63.4% and 79.8%, respectively. Additionally, treatment T11 involving farmyard manure + 75% RDN possessed maximum Zn, Cu, Fe and Mn concentrations as compared to the other treatments. Additionally, T11 was not statistically different from treatment T12 for Cu, Fe and Mn concentrations in grain. However, in straw yield, treatment T11 was not statistically different from treatment T12 for Zn, Cu and Fe concentrations, respectively.

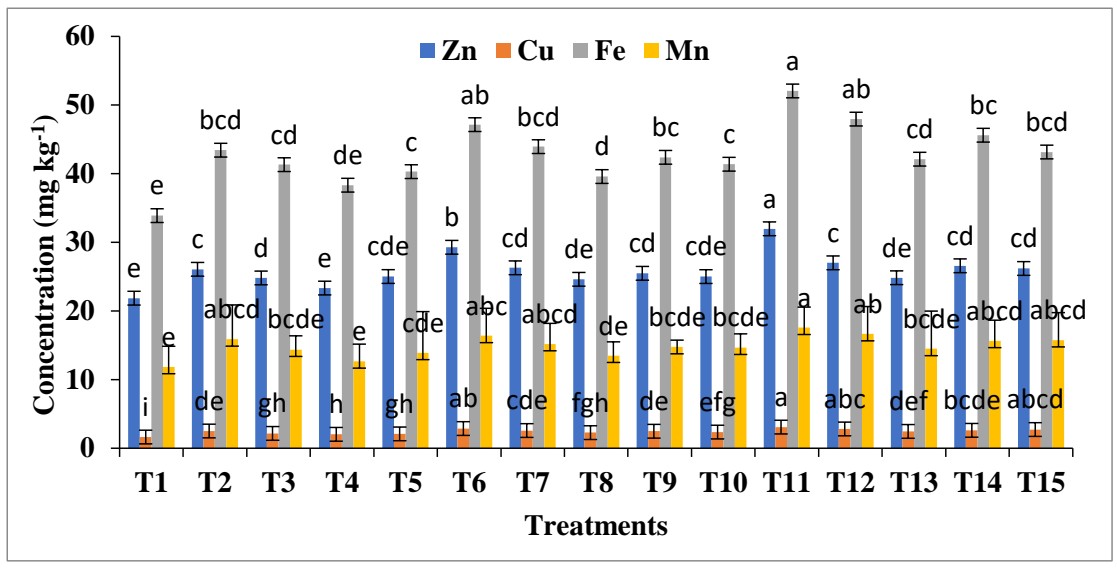

(**a**)

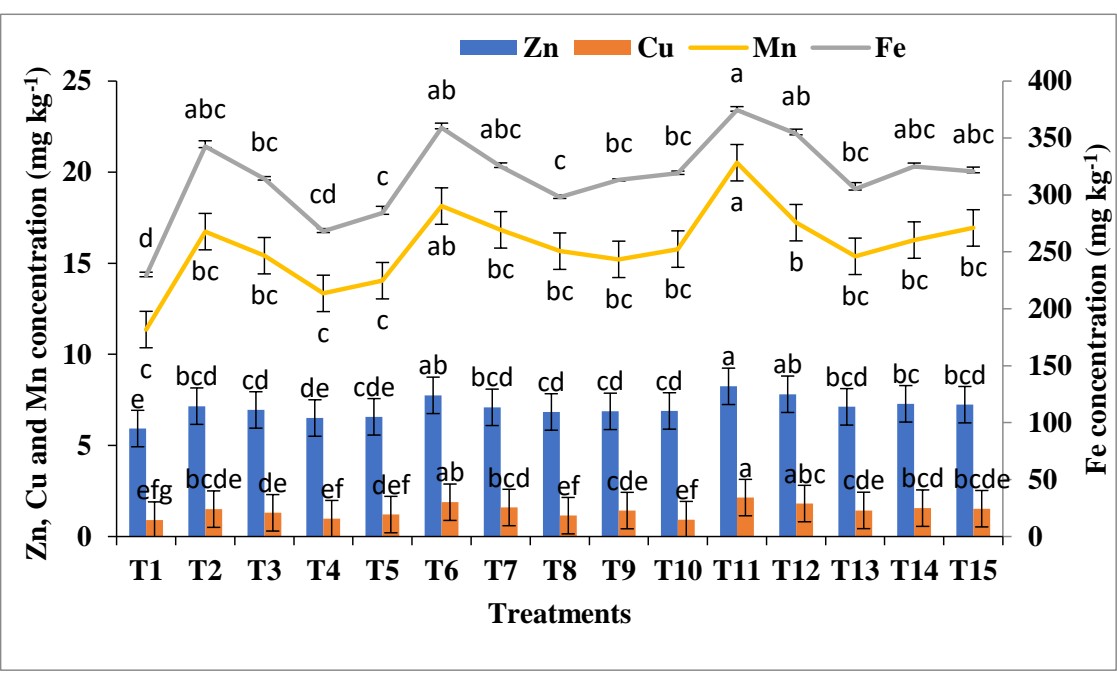

(**b**)

**Figure 3.** Effect of different fertilizers and manures on micronutrient concentration (Zn, Cu, Fe and Mn) in (**a**) grain and (**b**) straw of wheat. Treatment details are given in Table 1. The bar with similar or dissimilar letter(s) was evaluated with the least significant difference (LSD) multiple range tests using a probability level of $p \leq 0.05$.

With the application of farmyard manure + 75% RDN, there was a considerable rise in Zn concentration. This is due to gradual release of Zn from farmyard manure after mineralization, as well as its chelating effect, which ensures a steady supply of Zn [51]. The Fe content of wheat was also dramatically enhanced by using farmyard manure. Organic matter application, according to Hegde [52], resulted in a considerable enhancement in available Fe content. Fe content of crops increased with the addition of farmyard manure, along with 75% RDN, and this was attributed to the additional amount of nutrients supplied by farmyard manure, which increased the yield and nutrient content [53]. The lower Fe

content in the control plot could be attributed to the low availability of micronutrients due to continuous cropping. Additionally, increasing the N concentration in grain and straw also increased the level of minerals such as Mg and Fe, etc. [54]. Manganese (Mn) and Cu followed a similar pattern to Zn and Fe, with the highest Mn and Cu concentrations found in farmyard manure + 75% RDN treatment. The accumulation of organic matter in wheat due to continual manuring may result in a higher total micronutrient content. Swarup concluded that the combined use of manures resulted in the easy availability of natural and applied micronutrient cations (Zn, Fe and Mn) in crops [55]. These elements tend to form stable complexes with organic ligands that diminish their susceptibility to adsorption or precipitation in soil. Use of farmyard manure may have led to the development of such metal–organic complexes of higher availability.

### 3.5. Impact of Manures and Fertilizers Application on Macronutrients Uptake of Grain and Straw

The outcomes of the present experiment depicted that macronutrient uptake in grain and straw of wheat were enhanced significantly with the sole and combined use of manures and inorganic fertilizers (Table 4). In grain, the N, P and K uptake increased up to 82.9%, 101.8% and 190.0%, respectively. However, in straw, the N, P and K uptake varied from 191.5%, 169.2% and 102.4%, respectively. Moreover, treatments involving farmyard manure (T2, T6 and T11 treatments) recorded the highest macronutrient uptake in wheat grain and straw samples over other treatments. Additionally, treatment T11 possessed maximum N, P and K uptake values in the grain and in straw of wheat. Additionally, in the case of grain yield, T11 was not statistically different from treatment T12 for N and K uptake, respectively. However, in straw yield, treatment T11 was not statistically different from treatment T12 for P and K uptake, respectively.

**Table 4.** Effect of different fertilizers and manures on N, P and K macronutrients uptake in grain and straw of wheat.

| Treatments | Macronutrients Uptake (kg ha$^{-1}$) by Wheat | | | | | |
|:---:|:---:|:---:|:---:|:---:|:---:|:---:|
| | Grain | | | Straw | | |
| | N | P | K | N | P | K |
| $T_1$ | 4.03 [h] | 1.64 [j] | 1.70 [g] | 0.59 [h] | 0.39 [h] | 6.24 [g] |
| $T_2$ | 4.87 [efg] | 2.13 [fgh] | 2.31 [f] | 0.86 [efgh] | 0.62 [efg] | 8.42 [def] |
| $T_3$ | 4.69 [fg] | 2.02 [ghi] | 2.21 [f] | 0.83 [efgh] | 0.55 [fgh] | 8.06 [defg] |
| $T_4$ | 4.38 [gh] | 1.83 [ij] | 1.94 [fg] | 0.61 [gh] | 0.49 [gh] | 6.88 [fg] |
| $T_5$ | 4.53 [fgh] | 1.93 [hi] | 2.04 [fg] | 0.84 [efgh] | 0.54 [fgh] | 7.61 [efg] |
| $T_6$ | 5.52 [cd] | 2.57 [cd] | 3.39 [cd] | 1.06 [be] | 0.75 [cde] | 10.01 [b] |
| $T_7$ | 5.06 [de] | 2.38 [df] | 3.06 [de] | 0.95 [df] | 0.66 [defg] | 8.95 [cdef] |
| $T_8$ | 4.70 [fg] | 2.10 [gh] | 2.78 [e] | 0.73 [fg] | 0.56 [efgh] | 7.95 [deg] |
| $T_9$ | 5.17 [def] | 2.23 [efg] | 2.91 [e] | 0.93 [ef] | 0.63 [defg] | 8.55 [def] |
| $T_{10}$ | 5.37 [cde] | 2.28 [efg] | 3.17 [cde] | 0.89 [efg] | 0.66 [defg] | 8.99 [bcde] |
| $T_{11}$ | 7.37 [a] | 3.31 [a] | 4.93 [a] | 1.72 [a] | 1.05 [a] | 12.63 [a] |
| $T_{12}$ | 6.95 [ab] | 3.01 [b] | 4.52 [a] | 1.34 [b] | 0.93 [abc] | 11.06 [ab] |
| $T_{13}$ | 5.89 [c] | 2.48 [cde] | 3.53 [bc] | 1.05 [cd] | 0.70 [def] | 9.91 [bcd] |
| $T_{14}$ | 6.58 [b] | 2.71 [c] | 3.94 [b] | 1.25 [bc] | 0.82 [bcd] | 10.7 [abc] |
| $T_{15}$ | 5.93 [c] | 2.49 [cde] | 3.45 [cd] | 1.23 [bcd] | 1.01 [ab] | 10.8 [abc] |
| LSD ($p \leq 0.05$) | 0.64 | 0.26 | 0.45 | 0.28 | 0.19 | 2.09 |

Treatment details are given in Table 1. The values having identical superscript letter do not differ significantly at 5% level by Duncan's Multiple Range test. NS—nonsignificant.

The increase in macronutrient uptake noted in the present experiment might be due to the increased grain and straw yield, as well as the introduction of adequate amounts of nitrogen through organic manures. Nitrogen uptake by the plant is a good indicator of a crop's nitrogen utilization efficiency. When farmyard manure was coupled with a 75% RDN fertilizer, nitrogen absorption increased in the current study. The reason for this is that farmyards have a higher C:N ratio, which causes N to be immobilized [56], while sole application of urea leads to nitrogen loss due to leaching and volatilization [57,58]. Farmyard manure enhances soil characteristics, allowing for the establishment of a larger root density and reduced leaching, which increases the crop's nutrient absorption capability. These findings suggested that combining organic and inorganic sources to improve macronutrient absorption in wheat is more effective [47]. Improved soil physical conditions and increased water availability improved the crop's nutrient absorption capability, resulting in increased biological output at a given quantity of fertilizer application [59]. Furthermore, the combination of farmyard manure with NPK contributes to better urea N availability and farmyard manure water absorption characteristics, resulting in higher micronutrient uptake. Due to greater grain yield and nutrient content, a similar trend in P and K uptake was observed. These findings corroborate those of Islam et al. [1] and Akter et al. [60] who found that combining manures with fertilizers increased N uptake by wheat and rice.

### 3.6. Impact of Manures and Fertilizers Application on Micronutrients Uptake of Grain and Straw

Data for micronutrient uptake in wheat affected by manures and fertilizers are presented in Table 5. The results demonstrated that the micronutrient (Zn, Cu, Fe and Mn) uptake in grain and straw of wheat was enhanced with the sole and combined use of manures and inorganic fertilizers over the control. In grain, Zn, Cu, Fe and Mn uptake varied from 135.1%, 187.6%, 135.7% and 160.5%, respectively. However, in straw, the Zn, Cu, Fe and Mn uptake varied from 34.4 to 67.6 g ha$^{-1}$, 5.09 to 16.8 g ha$^{-1}$, 1340.4 to 3026.9 g ha$^{-1}$ and 39.1 to 89.6 g ha$^{-1}$, respectively. Moreover, the Zn and Fe uptake was found to be maximum in treatment T11 in grain and straw, respectively. In the case of Fe, the results were not statistically different from treatment T12 in wheat straw. In case of Cu, the treatment T11 was not statistically different from treatment T12 in wheat grain.

The higher uptake of micronutrients with the application of organic manures and inorganic fertilizers might be attributed to the higher bioavailability of micronutrients in the soil, which increased their uptake in wheat in comparison to control [61]. The breakdown of organic matter releases nutrients through various mechanisms, which increases their availability in soil [62]. Moreover, addition of organic manures leads to lowering in soil pH due to the release of various acids, which favor the enhanced bioavailability of micronutrients compared to the sole use of inorganic fertilizers. Similar results have been obtained by Dhaliwal et al. [63], in which higher uptakes of micronutrients have been recorded with combined use of inorganic and organic fertilizer-treated plots compared to control and sole use of inorganic fertilizers. Additionally, synergistic application of both organic and inorganic fertilizer may provide a sufficient level of nutrition, which takes part in the biosynthesis of chlorophyll. The micronutrient uptake was also influenced by their interactions with other nutrients. This interaction among nutritional components was further influenced by the physiochemical qualities of the soil [64]. The application of N and Zn has a positive relationship [65]. Additionally, high N concentrations result in an increase in the number of Zn transporters and Zn chelating nitrogenous substances [66]. More nitrogen fertilization increases Zn uptake and translocation in wheat roots and shoots by 300% [67], resulting in increased Zn uptake in grain [68]. Manganese also has a favorable interaction with N. Because nitrate has the ability to promote Mn availability, uptake of Mn in wheat rises when N fertilizers such as urea are applied [69,70]. Moreover, the use of N fertilizers as nitrate ($NO_3^-$) enhances the rhizospheric pH, whereas ammonium ions ($NH_4^+$) lead to a higher uptake of Mn. Likewise, Cu and Fe uptake was increased significantly due to the combined use of organic and inorganic fertilizers.

**Table 5.** Effect of different fertilizers and manures on Zn, Cu, Fe and Mn micronutrients uptake in grain and straw of wheat.

| Treatments | Micronutrients Uptake (g ha$^{-1}$) by Wheat | | | | | | | |
|---|---|---|---|---|---|---|---|---|
| | **Grain** | | | | **Straw** | | | |
| | **Zn** | **Cu** | **Fe** | **Mn** | **Zn** | **Cu** | **Fe** | **Mn** |
| T$_1$ | 73.7 h | 5.39 h | 111.9 i | 65.5 f | 34.4 g | 5.09 g | 1340.4 f | 39.1 g |
| T$_2$ | 96.4 fg | 9.40 def | 158.5 efh | 115.9 bcd | 46.9 cdef | 9.82 de | 2252.1 bcde | 58.9 de |
| T$_3$ | 95.3 fg | 7.88 fg | 145.6 gh | 100.3 ce | 42.9 efg | 8.28 def | 1999.4 de | 52.7 ef |
| T$_4$ | 83.3 gh | 7.27 g | 135.5 hi | 79.3 ef | 39.2 fg | 5.99 fg | 1654.3 ef | 45.0 fg |
| T$_5$ | 97.4 f | 7.53 g | 142.6 gh | 86.8 def | 39.8 fg | 7.43 efg | 1837.7 ef | 50.4 ef |
| T$_6$ | 130.5 bc | 11.7 bc | 192.3 cd | 128.5 bc | 51.1 bcde | 12.9 bc | 2652.8 abc | 68.8 c |
| T$_7$ | 100.1 ef | 10.0 cde | 171.5 def | 113.5 bcd | 45.9 cdef | 10.2 cd | 2281.7 bcde | 59.3 de |
| T$_8$ | 94.8 fg | 8.58 efg | 146.0 gh | 100.1 cde | 43.2 def | 7.57 defg | 1896.5 def | 50.2 ef |
| T$_9$ | 105.7 ef | 9.68 de | 156.3 fgh | 103.4 cde | 44.5 def | 9.39 de | 2091.9 cde | 57.1 de |
| T$_{10}$ | 109.3 def | 9.65 de | 165.4 efg | 108.0 bcde | 46.5 cdef | 6.37 fg | 2133.9 cde | 57.8 de |
| T$_{11}$ | 173.3 a | 15.5 a | 263.8 a | 170.6 a | 67.6 a | 16.8 a | 3026.9 a | 89.6 a |
| T$_{12}$ | 129.3 bc | 13.5 a | 231.6 b | 134.6 b | 59.5 ab | 13.3 b | 2787.4 ab | 80.5 ab |
| T$_{13}$ | 114.4 cde | 10.8 bcd | 179.8 def | 107.0 bde | 48.4 bcde | 9.74 de | 2246.7 bcde | 59.7 de |
| T$_{14}$ | 138.4 b | 12.4 ab | 207.8 bc | 128.4 bc | 54.2 bcd | 11.3 bd | 2471.7 abcd | 71.5 bc |
| T$_{15}$ | 124.5 bcd | 11.6 bc | 182.2 de | 138.9 ab | 56.2 bc | 10.9 bc | 2530.0 abcd | 66.8 cd |
| LSD ($p \leq 0.05$) | 16.1 | 1.7 | 24.9 | 65.5 f | 16.1 | 2.7 | 633.5 | 11.1 |

Treatment details are given in Table 1. The values with identical superscript letters do not differ significantly at 5% level by Duncan's Multiple Range test. NS—non significant.

### 3.7. Effect of Manures and Fertilizers Application on Soil Properties

The addition of organic manures increased soil EC due to the increase in soluble salt concentration sourced from feed additives. The decomposition of manures releases acids which chemically react with sparingly soluble salts and increase solubility. The increase in OC was due to the addition of organic matter through manures. The results recorded a significant increase in NPK content with organic manures and fertilizers (Table 6). The lower macronutrient status in the control treatment was attributed to the mining of available macronutrients by crops without fertilization for two years. The improved macronutrient levels in manure-added plots could probably be attributed to the nutrient supplemented to the soil from the decomposition of organic matter and release of $CO_2$. In the case of P, the organic acids released during the decomposition of organic manures accelerate the P solubilization due to enhanced microbial activities and due to retarded P fixation in soil. The higher K content in the soil might be associated with the increased cation exchange capacity of soil, which improved the capacity of soil to hold more exchangeable K and reduced leaching losses of K in soil [71]. In NPK-incorporated plants, the nutrients were added externally to the soil, thus there was an increase in soil nutrient level. These results corroborate the findings of previous studies, in which the addition of manures and inorganic fertilizers has improved the nutrient status of soil [72].

**Table 6.** Interactive effect of manures and fertilizers on soil properties after the harvesting of wheat crop, 2021.

| Treatments | pH | EC (dS m$^{-1}$) | OC (%) | N (kg ha$^{-1}$) | P (kg ha$^{-1}$) | K (kg ha$^{-1}$) |
|---|---|---|---|---|---|---|
| T1 | 7.68 | 0.27 [f] | 0.35 [f] | 152 [g] | 27.67 [i] | 79.4 [g] |
| T2 | 7.80 | 0.31 [de] | 0.43 [cd] | 190 [cd] | 34.87 [h] | 140.6 [a] |
| T3 | 7.95 | 0.31 [de] | 0.39 [def] | 169 [ef] | 36.08 [h] | 100.5 [e] |
| T4 | 7.75 | 0.29 [ef] | 0.46 [b] | 201 [bc] | 41.87 [ef] | 97.6 [f] |
| T5 | 7.64 | 0.29 [ef] | 0.38 [ef] | 166 [f] | 44.00 [cdef] | 106.4 [def] |
| T6 | 7.53 | 0.28 [ef] | 0.52 [a] | 228 [a] | 40.90 [fg] | 114.7 [cd] |
| T7 | 7.98 | 0.33 [c] | 0.41 [de] | 179 [d] | 45.78 [abcd] | 119.3 [bc] |
| T8 | 7.64 | 0.36 [abc] | 0.49 [ab] | 216 [ab] | 47.43 [abc] | 107.8 [de] |
| T9 | 7.65 | 0.27 [f] | 0.42 [cde] | 184 [de] | 34.73 [h] | 103.2 [ef] |
| T10 | 7.68 | 0.30 [ef] | 0.38 [ef] | 166 [f] | 45.42 [bcde] | 118.0 [b] |
| T11 | 7.53 | 0.39 [a] | 0.51 [a] | 225 [a] | 48.10 [ab] | 143.0 [a] |
| T12 | 7.78 | 0.33 [c] | 0.46 [bc] | 201 [bc] | 42.94 [def] | 103.9 [ef] |
| T13 | 7.62 | 0.35 [bc] | 0.49 [ab] | 216 [ab] | 49.08 [a] | 126.3 [b] |
| T14 | 7.59 | 0.31 [de] | 0.44 [c] | 193 [c] | 38.09 [gh] | 102.7 [ef] |
| T15 | 7.60 | 0.37 [ab] | 0.39 [def] | 171 [ef] | 42.00 [ef] | 101.3 [ef] |
| LSD ($p \leq 0.05$) | NS | 0.03 | 0.04 | 15 | 3.58 | 9.8 |

Treatment details are given in Table 1. In the column, mean with similar or dissimilar letter(s) was evaluated with the least significant difference (LSD) multiple range tests using a probability level of $p \leq 0.05$.

## 4. Conclusions

The present study indicates that the partial substitution of inorganic fertilizers with organic manures significantly improved the plant growth parameters, crop yield, soil properties and nutrient build-up in a basmati rice–wheat cropping system. The addition of farmyard manure along with 75% RDN showed superior results in terms of crop yield and macronutrient uptake. Similar results were observed for zinc, copper, iron and manganese uptake. The results of the treatment involving farmyard manure were found to be on par with the treatment involving poultry manure. The combined use of farmyard manure with 75% RDN (T11) improved the grain and straw yield by 33.6% and 34.9%, respectively compared to control. The presence of high N content in farmyard manure favored the vegetative growth as well as improved the nutrient status of the crop. The residual effect of manures before wheat transplantation increased the wheat yield, which ultimately enhanced the concentration and the accumulation of macro- and micro-nutrients in comparison to the treatments without manures. Thus, the combined application of manures with chemical fertilizers over the sole application of inorganic fertilizers provides essential nutrients to the crop, which could be considered as an alternate way of enhancing nutritional security and sustaining crop productivity. In the present study, the main emphasis has been given to recommended levels of N only by ignoring the other nutrients in different manures. Further evaluation pertaining to the interactive effect of manures and fertilizers on soil fertility build-up is required with respect to soil micronutrients and soil biochemical properties.

**Author Contributions:** S.S.D. and V.S.: Conceptualization, funding acquisition, methodology, writing—original draft; V.S. and R.K.G.: Investigation, formal analysis; A.K.S.: Investigation, methodology, supervision, validation, writing—original draft; V.V.: Software, investigation; M.K.: Investigation, methodology, S.K.B.: Methodology, investigation, writing—original draft; P.S.: Formal analysis, investigation, methodology. All authors have read and agreed to the published version of the manuscript.

**Funding:** The authors are thankful to the Indian Council of Agricultural Research, New Delhi for extending financial support through All India Coordinated Research Project on Micro and Secondary Nutrients and Pollutant Elements in Soils and Plants to carry out the present study.

**Institutional Review Board Statement:** Not applicable.

**Informed Consent Statement:** Not applicable.

**Data Availability Statement:** The data presented in this study are available on request from the authors.

**Acknowledgments:** The authors thank the Head of the Department of Soil Science, Punjab Agricultural University, Ludhiana, 141004, India for providing the research facilities to carry out the work.

**Conflicts of Interest:** The authors declare no conflict of interest.

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
