# Peer review of "Residual Effect of Organic and Inorganic Fertilizers on Growth, Yield and Nutrient Uptake in Wheat under a Basmati Rice–Wheat Cropping System in North-Western India"

_agriculture, doi:10.3390/agriculture13030556_

Round 1
Reviewer 1 Report
General comments
-This manuscript, by authors, studied “
“Residual effect of organic and inorganic fertilizers on growth, yield and nutrient uptake in wheat under basmati-wheat cropping system in North-Western India”.
Overall, the topic is of interest to Agriculture, readers. However, the following are the specific comments on the article concerns, before publication as major revision.
- All figures and tables need to improve.
- Figures need analysis for significant difference
Specific Comments and Suggestions
-Abstract
-Add more results in detail.
-Experiment design is not clear
-Significance of your study? Specific findings?
- Add quantitative results.
-Introduction
-Need to summarize and be specific with your concerned study.
Add more details about the relationship between organic fertilization and greenhouse gasses emissions. May consider this and more related
“Reuse of agricultural wastes, manure, and biochar as an organic amendment: A review on its implications for vermicomposting technology”
-Revise the introduction section with summarize and significance of the study.
-Mention specific objectives of your study
-Can add more references.
- What about crop rotation in long-term field experiments.
-Materials and Methods
-Methodology also needs to summarize and be specific.
-Must need to improve experimental design.
-15 treatments are too many and cannot be described clearly.
-What is the baseline of NPK dose?
-On which basis has statistical analysis been performed? as doses are not same.
-Add more details about soil and its importance.
-Plot treatments and their doses can be more clear. A table can provide better details.
- Please add experimental setup (field and lab experiments), pictures, and more figures to make the manuscript more attractive.
-Results and Discussion
-A weak discussion, please improve accordingly. Please add other studies' references to support your current study and its significance
-Conclusions
-More specific, please?
-Significant differences on which basis?
Author Response
We sincerely thank editor and reviewers for the valuable comments, which are greatly appreciated. In the revised manuscript, we have highlighted the changes in track change mode. The point wise replies are as below.
Comments from the Editor and Reviewers:
Reviewer 1
General comments
-This manuscript, by authors, studied “Residual effect of organic and inorganic fertilizers on growth, yield and nutrient uptake in wheat under basmati wheat cropping system in North-Western India”. Overall, the topic is of interest to Agriculture, readers. However, the following are the specific comments on the article concerns, before publication as major revision.
Comment: All figures and tables need to improve.
Reply: All the figures and tables have been improved as suggested.
Comment: Figures need analysis for significant difference
Reply: Analysis of significant difference has been added in the figures displaying concentration data for macro and micronutrients.
Specific Comments and Suggestions
Abstract
Comment: Add more results in detail.
Reply: More results have been added in the abstract.
Comment: Experiment design is not clear.
Reply: Experimental details have been added in the abstract.
Comment: Significance of your study? Specific findings?
Reply: Specific finding have been added.
Comment: Add quantitative results.
Reply: More quantitative data has been added.
Introduction
Comment: Need to summarize and be specific with your concerned study.
Reply: Introduction has been rephrased and summarized as suggested.
Comment: Add more details about the relationship between organic fertilization and greenhouse gasses emissions. May consider this and more related
“Reuse of agricultural wastes, manure, and biochar as an organic amendment: A review on its implications for vermicomposting technology”
Reply: Relationship between organic fertilizers and greenhouse gas emission has been mentioned in the introduction section.
Comment: Revise the introduction section with summarize and significance of the study.
Reply: Introduction has been rephrased.
Comment: Mention specific objectives of your study.
Reply: Objectives have been mentioned in the introduction.
Comment: Can add more references.
Reply: More references have been added in the introduction section.
Comment: What about crop rotation in long-term field experiments.
Reply: The results of long-term field experiment conducted by other researchers have been discussed in introduction section.
Materials and Methods
Comment: Methodology also needs to summarize and be specific.
Reply: Methodology has been rephrased as suggested.
Comment: Must need to improve experimental design.
Reply: Experimental design has been improved.
Comment: 15 treatments are too many and cannot be described clearly.
Reply: The specific objective of the present study was to determine the most effective combination of organic manures and inorganic fertilizer to improve yield and nutrient uptake in wheat due to this different combinations were studied in 15 treatments.
Comment: What is the baseline of NPK dose?
Reply: Recommended dose of nitrogen was used according to the package of practices, PAU, Ludhiana.
Comment: On which basis has statistical analysis been performed? as doses are not same.
Reply: One way analysis of variance (ANOVA) and Duncan Multiple Range test was performed in order to assess significant difference between the treatment results on the crop.
Comment: Add more details about soil and its importance.
Reply: Different parameters of soil have been added in the material and method section.
Comment: Plot treatments and their doses can be more clear. A table can provide better details.
Reply: Detailed description of organic manures and different dose of RDN have been added in table 1.
Comment: Please add experimental setup (field and lab experiments), pictures, and more figures to make the manuscript more attractive.
Reply: Detailed experimental setup has been given in material and method section.
Results and Discussion
Comment: A weak discussion, please improve accordingly. Please add other studies' references to support your current study and its significance.
Reply: More discussion has been added along with relevant references.
Conclusions
Comment: More specific, please?
Reply: Conclusion has been rephrased.
Reviewer 2
This paper introduces the effects of organic fertilizer and chemical fertilizer on wheat growth and yield, but there are many similar studies which did not show a certain innovation.
Comment: There are some problems in the experimental design of this paper. For example, in the 15 experimental treatments, it is not explained whether the contents of N, P and K of fertilizers used in each treatment are similar or not, and better growth results can be obtained naturally with high fertilization nutrients.
Reply: Experimental design has been improved. Recommended dose of nitrogen was used as an inorganic fertilizer for the present experiment.
Comment: In different experimental treatments, the soil fertility and basic physical and
chemical properties of each treatment soil should have analytical data or descriptions;
Reply: Different soil properties have been added in material and method section.
Comment: Thirdly, the method of collecting experimental samples for nutrient analysis is not fully written.
Reply: The method used for nutrient analysis has been fully written in material and method section.
Comment: Because of the above problems, it is difficult to determine the reliability of the research results. I suggest that the relevant data of manuscript may be supplemented by the author.
Reply: Material and method has been improved.
Reviewer 3
The work deals with interesting issues regarding the different of organic and inorganic fertilizers two years’ fertilization regimes and its effect on wheat productivity changes. The work mainly concerns issues in the field of agronomy. Study confirms that the findings could have certain practical significance for improving combined application of manures with chemical fertilizers providing essential nutrients could be considered as an alternate way of enhancing nutritional security and sustaining crop productivity,
Thanks for appreciating the work.
However, I have a few comments for further improvement as
follows:
Comment: In Materials and Methods section, it is worth presenting the field plan of the experimental research in graphical expression or graphical abstract, to make it easier to understand whole experiment.
Reply: Experimental design has been presented in the tabular form in material and method section.
Comment: Meteorological conditions is quite important for agronomical research. Rainfall quantities were presented, but it would be useful to provide the monthly temperature and other parameters (Line 83).
Reply: Data pertaining to the monthly temperature and other parameters for two years i.e. 2019-2020 and 2020-2021 has been presented in figure 1.
Comment: There is some unclear markings, like @ (Line 89) and in other text parts.
Reply: Marking @ has been removed.
Comment: It would be useful to explain why the respective rates of fertilization with different manures were chosen, e.g. for Farmyard manure (15 t ha-1), for Poultry manure (6 t ha-1) and so on. Is it because due to nitrogen leaching or other reasons.
Reply: These are the recommended doses.
Comment: Dimension mg Kg should be changed to mg kg (Line 224) and in other text parts.
Reply: Dimensions have been changed in text and figures.
Comment: Conclusions have to contain purposes and suggestions, according with introduction, abstract and title of the paper as well. The conclusions should respond to the statements of the set main objective. In my opinion, the conclusions should be more concrete and specific comparable quantitative results could be use. According to conclusions Farmyard manure with 75% NPK (T11) variant is the most optimal. Perhaps it is worth mentioning other variants, or which of them are not recommended.
Reply: Conclusion has been rephrased as suggested.
Comment: List of the references are not prepared in accordance with the requirements. There some formatting errors.
Reply: References have been written following the author guidelines.

Reviewer 2 Report
This paper introduces the effects of organic fertilizer and chemical fertilizer on wheat growth and yield, but there are many similar studies,which did not show a certain innovation.
There are some problems in the experimental design of this paper. For example, in the 15 experimental treatments, it is not explained whether the contents of N, P and K of fertilizers used in each treatment are similar or not, and better growth results can be obtained naturally with high fertilization nutrients. In different experimental treatments, the soil fertility and basic physical and chemical properties of each treatment soil should have analytical data or descriptions; Thirdly, the method of collecting experimental samples for nutrient analysis is not fully written.
Because of the above problems, it is difficult to determine the reliability of the research results.I suggest that the relevant data of manuscript may be supplemented by the author.
Author Response

(The authors gave the same response as above.)

Reviewer 3 Report
Comments and Suggestions for Authors
The work deals with interesting issues regarding the different of organic and inorganic fertilizers two years’ fertilization regimes and its effect on wheat productivity changes. The work mainly concerns issues in the field of agronomy. Study confirms that the findings could have certain practical significance for improving combined application of manures with chemical fertilizers providing essential nutrients could be considered as an alternate way of enhancing nutritional security and sustaining crop productivity,
However, I have a few comments for further improvement as follows:
1 1. In Materials and Methods section, it is worth presenting the field plan of the experimental research in graphical expression or graphical abstract, to make it easier to understand whole experiment.
2. Meteorological conditions is quite important for agronomical research. Rainfall quantities were presented, but it would be useful to provide the monthly temperature and other parameters (Line 83).
3 3. There is some unclear markings, like @ (Line 89) and in other text parts.
4. It would be useful to explain why the respective rates of fertilization with different manures were chosen, e.g. for Farmyard manure (15 t ha-1), for Poultry manure (6 t ha-1) and so on. Is it because due to nitrogen leaching or other reasons.
5. Dimension mg Kg-1 should be changed to mg kg-1 (Line 224) and in other text parts.
6. Conclusions have to contain purposes and suggestions, according with introduction, abstract and title of the paper as well. The conclusions should respond to the statements of the set main objective. In my opinion, the conclusions should be more concrete and specific comparable quantitative results could be use. According to conclusions Farmyard manure with 75% NPK (T11) variant is the most optimal. Perhaps it is worth mentioning other variants, or which of them are not recommended.
7. List of the references are not prepared in accordance with the requirements. There some formatting errors.
Those and other comments are provided in the manuscript.
Author Response

(The authors gave the same response as above.)

Round 2
Reviewer 2 Report
The revised version has been greatly improved, but some experimental design and other contents cannot be improved anymore. It is suggested that the author point out the limitations of this paper in experimental design in the conclusion (according to my comments last time).
Author Response
We sincerely thank editor and reviewers for the valuable comments, which are greatly appreciated. In the revised manuscript, we have highlighted the changes in track change mode. The point wise replies are as below.
Comments from the Editor and Reviewers:
Comment: The revised version has been greatly improved, but some experimental design and other contents cannot be improved anymore. It is suggested that the author point out the limitations of this paper in experimental design in the conclusion (according to my comments last time).
Reply: Thanks for appreciating the work. The paper along with treatment details has been revised as per the suggestions. The data pertaining to effect of manures and fertilizers on soil properties has been incorporated. The limitations of the study have been added in the conclusion section as suggested.
Regards
Arvind Kumar Shukla
